# Canonical 3D Deformer Maps:
# Unifying parametric and non-parametric methods for dense weakly-supervised category reconstruction

**David Novotny**\*    **Roman Shapovalov**\*    **Andrea Vedaldi**
Facebook AI Research
{dnovotny, romansh, vedaldi}@fb.com
http://www.robots.ox.ac.uk/~david/c3dm/

## Abstract

We propose the *Canonical 3D Deformer Map*, a new representation of the 3D shape of common object categories that can be learned from a collection of 2D images of independent objects. Our method builds in a novel way on concepts from parametric deformation models, non-parametric 3D reconstruction, and canonical embeddings, combining their individual advantages. In particular, it learns to associate each image pixel with a deformation model of the corresponding 3D object point which is canonical, i.e. intrinsic to the identity of the point and shared across objects of the category. The result is a method that, given only sparse 2D supervision at training time, can, at test time, reconstruct the 3D shape and texture of objects from single views, while establishing meaningful dense correspondences between object instances. It also achieves state-of-the-art results in dense 3D reconstruction on public in-the-wild datasets of faces, cars, and birds.

## 1   Introduction

We address the problem of learning to reconstruct 3D objects from individual 2D images. While 3D reconstruction has been studied extensively since the beginning of computer vision research [49], and despite exciting progress in monocular reconstruction for objects such as humans, a solution to the general problem is still elusive. A key challenge is to develop a *representation* that can learn the 3D shapes of common objects such as cars, birds and humans from 2D images, without access to 3D ground truth, which is difficult to obtain in general. In order to do so, it is not enough to model individual 3D shapes; instead, the representation must also *relate the different shapes* obtained when the object deforms (e.g. due to articulation) or when different objects of the same type are considered (e.g. different birds). This requires establishing *dense correspondences* between different shapes, thus identifying equivalent points (e.g. the left eye in two birds). Only by doing so, in fact, the problem of reconstructing independent 3D shapes from 2D images, which is ill-posed, reduces to learning a single deformable shape, which is difficult but approachable.

In this paper, we introduce the *Canonical 3D Deformer Map* (C3DM), a representation that meets these requirements (Figure 1). C3DM combines the benefits of parametric and non-parametric representations of 3D objects. Conceptually, C3DM starts from a *parametric 3D shape model* of the object, as often used in Non-Rigid Structure From Motion (NR-SFM [11]). It usually takes the form of a *mesh* with 3D vertices $\mathbf{X}_1, \ldots, \mathbf{X}_K \in \mathbb{R}^3$ expressed as a linear function of global deformation parameters $\boldsymbol{\alpha}$, such that $\mathbf{X}_k = B_k \boldsymbol{\alpha}$ for a fixed operator $B_k$. Correspondences between shapes are captured by the identities $k$ of the vertices, which are invariant to deformations. Recent works such as Category-specific Mesh Reconstruction (CMR) [31] put this approach on deep-learning rails,

---

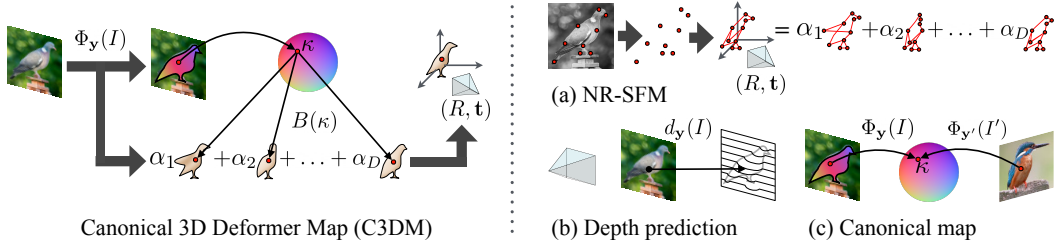

Canonical 3D Deformer Map (C3DM)       (a) NR-SFM      (b) Depth prediction      (c) Canonical map

Figure 1: The C3DM representation (left) associates each pixel $\mathbf{y}$ of the image $I$ with a *deformation operator* $B(\kappa)$, a function of the object canonical coordinates $\kappa = \Phi_{\mathbf{y}}(I)$. C3DM then reconstructs the corresponding 3D point $\mathbf{X}$ as a function of the global object deformation $\boldsymbol{\alpha}$ and viewpoint $(R, \mathbf{t})$. It extends three ideas (right): (a) non-rigid structure from motion computes a *sparse* parametric reconstruction starting from 2D keypoints rather than an image; (b) a monocular depth predictor $d_{\mathbf{y}}(I)$ non-parametrically maps each pixel to its 3D reconstruction but lacks any notion of correspondence; (c) a canonical mapping $\Phi_{\mathbf{y}}(I)$ establishes dense correspondences but does not capture geometry.

learning to map an image $I$ to the deformation parameters $\boldsymbol{\alpha}(I)$. However, working with meshes causes a few significant challenges, including guaranteeing that the mesh does not fold, rendering the mesh onto the image for learning, and dealing with the finite mesh resolution. It is interesting to compare parametric approaches such as CMR to *non-parametric depth estimation models*, which directly map each pixel $\mathbf{y}$ to a depth value $d_{\mathbf{y}}(I)$ [70, 33, 18], describing the geometry of the scene in a dense manner. The depth estimator $d_{\mathbf{y}}(I)$ is easily implemented by means of a convolutional neural network and is not bound to a fixed mesh resolution. However, a depth estimator has no notion of correspondences and thus of object deformations.

Our intuition is that these two ways of representing geometry, parametric and non-parametric, can be combined by making use of the third notion, a *canonical map* [57, 51, 35]. A canonical map is a non-parametric model $\Phi_{\mathbf{y}}(I) = \kappa$ that associates each pixel $\mathbf{y}$ to the intrinsic coordinates $\kappa$ of the corresponding object point. The latter can be thought of as a continuous generalization of the index $k$ that in parametric models identifies a vertex of a mesh. Our insight is that *any* intrinsic quantity — i.e. one that depends only on the identity of the object point — can then be written as a function of $\kappa$. This includes the *3D deformation operator* $B_{\kappa}$, so that we can reconstruct the 3D point found at pixel $\mathbf{y}$ as $\mathbf{X}_{\mathbf{y}} = B_{\kappa}\boldsymbol{\alpha}$. Note that this also requires to learn the mapping $\kappa \mapsto B_{\kappa}$, which we can do by means of a small neural network.

We show that the resulting representation, C3DM, can reconstruct the shape of 3D objects densely and from single images, using only easily-obtainable 2D supervision at training time — the latter being particularly useful for 3D reconstruction from traditional non-video datasets. We extensively evaluate C3DM and compare it to CMR [31], state-of-the-art method for monocular category reconstruction. C3DM achieves both higher 3D reconstruction accuracy and more realistic visual reconstruction on real-world datasets of birds, human faces, and four other deformable categories of rigid objects.

## 2 Related work

The literature contains many impressive results on image-based 3D reconstruction. To appreciate our contribution, it is essential to characterize the assumptions behind each method, the input they require for training, and the output they produce. Multiple works [41, 6, 21, 53, 10, 39, 27, 71, 29, 47, 67, 55, 61, 45, 46, 30, 34, 53, 39, 50, 46, 62, 47] take as input an existing parametric 3D model of the deformable object such as SMPL [41] or SCAPE [6] for humans bodies, or Basel [48] for faces and *fit it to images*. In our case, no prior parametric 3D model is available; instead, our algorithm *simultaneously learns and fits a 3D model using only 2D data as input*.

**Sparse NR-SFM** methods receive sparse 2D keypoints as input and lift them in 3D, whereas C3DM receives as input an image and produces a *dense* reconstruction. In other words, we wish to obtain dense reconstruction of the objects although only sparse 2D annotations are still provided during training. For learning, NR-SFM methods need to separate the effect of viewpoint changes and deformations [69]. They acheive it by constraining the space of deformations in one of the following ways: assume that shapes span a low-rank subspace [3, 17, 16, 76] or that 3D trajectories are smooth in time [4, 5], or combine both types of constraints [1, 19, 38, 37], or use multiple subspaces [76, 2],

sparsity [73, 74] or Gaussian priors [59]. In Section 3.2, we use NR-SFM to define one of the loss functions. We chose to use the recent C3DPO method [44], which achieves that separation by training a canonicalization network, due to its state-of-the-art performance.

**Dense 3D reconstruction.** Differently from our work, most of the existing approaches to dense 3D reconstruction assume either 3D supervision or rigid objects and multiple views. Traditional *multi-view* approaches [7] perform 3D reconstruction by analyzing disparities between two or more calibrated views of a rigid object (or a non-rigid object simultaneously captured by multiple cameras), but may fail to reconstruct texture-less image regions. Learning multi-view depth estimators with [70] or without [33] depth supervision can compensate for lacking visual evidence. The method of Innmann et al. [28] can reconstruct mildly non-rigid objects, but still requires multiple views.

Most methods for *single-view* dense reconstruction of object categories require 3D supervision [42, 56, 20]. In particular, AtlasNet [20] uses a representation similar to ours, mapping points on a two-dimensional manifold to points on the object's surface with a multi-layer perceptron (MLP). Instead of conditioning the MLP on a shape code, we map the manifold points to embeddings; the 3D location is defined as their linear combination. Only a few methods, like C3DM, manage to learn parametric shape from 2D data only: Cashman and Fitzgibbon [13] propose a morphable model of dolphins supervised with 2D keypoints and segmentation masks, while others [63, 12] reconstruct the categories of PASCAL VOC. Most of these methods start by running a traditional SFM pipeline to obtain the mean 3D reconstruction and camera matrices. Kar et al. [32] replace it with NR-SFM for reconstructing categories of PASCAL3D+. VpDR [43] reconstructs *rigid* categories from monocular views. Wu et al. [66] reconstruct non-rigid symmetrical shapes by rendering predicted depth maps, albedo, and shading and ensuring symmetry, which works well for limited viewpoint variation. In this work, C3DM reconstructs 3D shape from a single image without assuming symmetry, limited range of viewpoints, or images related by rigid transform at training or prediction time.

A number of recent methods based on *differentiable mesh rendering* can also be trained with 2D supervision only. Kanazawa et al. [31] introduced CMR, a deep network that reconstructs shape and texture of deformable objects; it is the closest to our work in terms of assumptions, type of supervision, and output, and is currently state of the art for reconstruction of categories other than humans. DIB-R [15] improves the rendering technique by softly assigning all image pixels, including background, to the mesh faces. Working with meshes is challenging since the model should learn to generate only valid meshes, e.g. those without face intersections. Henderson et al. [26] proposed parametrisation of a mesh that prevents intersecting faces. In contrast to these methods, we work with point clouds and avoid computationally expensive rendering by leveraging NR-SFM pre-processing and cross-image consistency constraints. The concurrent work, Implicit Mesh Reconstruction [60], defines similar constraints to our reprojection and cross-image consistency using rendering-like interpolation of 3D points on the mesh surface. We avoid this step by predicting 3D coordinates of each pixel in a feed-forward manner. IMR does not build, as we do, on NR-SFM. The advantage is that this enables a variant that trains without keypoint supervision. The disadvantage is that, in order do to so, IMR has to initialise the model with a hand-crafted category-specific template mesh.

**Canonical maps.** A *canonical map* is a function that maps image pixels to identifiers of the corresponding object points. Examples include the UV surface coordinates used by Dense Pose [22] and spherical coordinates [57]. Thewlis et al. [57, 58], Schmidt et al. [51] learn canonical maps in an unsupervised manner via a bottleneck, whereas Kulkarni et al. [35, 36] do so by using consistency with an initial 3D model. Normalized Object Coordinate Space (NOCS) [65] also ties canonical coordinates and object pose, however it does not allow for shape deformation; different shapes within category have to be modelled by matching to one of the hand-crafted exemplars. Instead, we learn the dense parametric 3D deformation model for each object category from 2D data.

## 3 Canonical 3D Deformer Map representation

### 3.1 The model

**Canonical map.** Let $I \in \mathbb{R}^{3 \times H \times W}$ be an image and $\Omega \subset \{1, \ldots, H\} \times \{1, \ldots, W\}$ be the image region that contains the object of interest. We consider a *canonical map* $\kappa = \Phi(\mathbf{y}; I)$ sending pixels $\mathbf{y} \in \Omega$ to points on the unit sphere $\kappa \in \mathbb{S}^2$, which is topologically equivalent to any 3D surface $\mathcal{S} \subset \mathbb{R}^3$ without holes. It can be interpreted as a space of indices or coordinates $\kappa$ that identify a dense

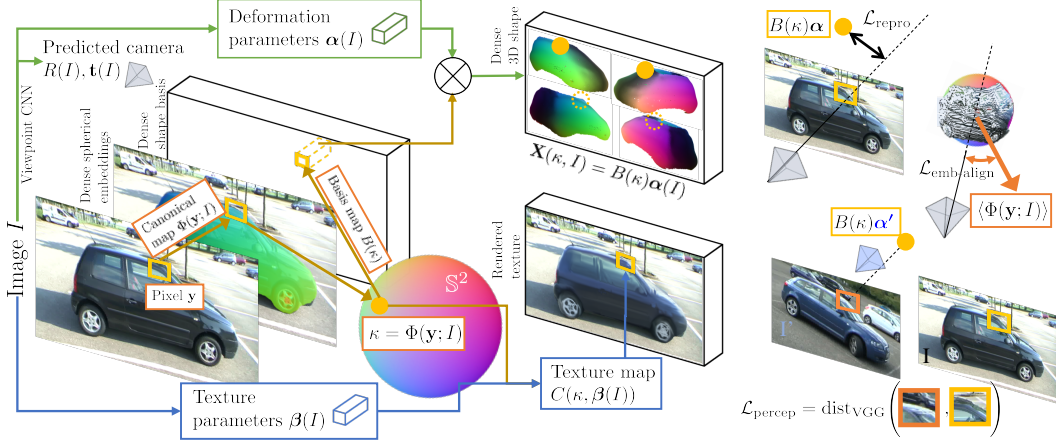

Figure 2: **Detailed system overview.** At test time, the image is passed through the network $\Phi$ to obtain the map of dense embeddings $\kappa \in \mathbb{S}^2$. The network $B$ converts them individually to deformation operators. In the meantime, the image is passed to the viewpoint network to predict the camera orientation $R$ and shape parameters $\boldsymbol{\alpha}$. Eq. (1) combines these quantities to obtain 3D reconstruction for each pixel within the object mask. At training time, sparse 2D keypoints are preprocessed with C3DPO [44] to obtain "ground truth" camera orientation $R^*$ and shape parameters $\boldsymbol{\alpha}^*$. These, together with the C3DPO basis $B^*$, are used in (4) to supervise the corresponding predicted variables. On the right, three more loss functions are illustrated: reprojection loss (5), cross-projection perceptual loss (6), and (8) aligning the camera orientation with average embedding direction.

system of 'landmarks' for the deformable object category. A landmark, such as the corner of the left eye in a human, is a point that can be identified repeatably despite object deformations. Note that the index space can take other forms than $\mathbb{S}^2$, however the latter is homeomorphic to most surfaces of 3D objects and has the minimum dimensionality, which makes it a handy choice in practice.

**Deformation model.** We express the 3D location of a landmark $\kappa$ as $\mathbf{X}(\kappa; I) = B(\kappa)\boldsymbol{\alpha}(I)$, where $\alpha(I) \in \mathbb{R}^D$ are image-dependent deformation parameters and $B(\kappa) \in \mathbb{R}^{3 \times D}$ is a linear operator indexed by $\kappa$. This makes $B(\kappa)$ an *intrinsic* property, invariant to the object deformation or viewpoint change. The full 3D reconstruction $\mathcal{S}$ is given by the image of this map: $\mathcal{S}(I) = \{B(\kappa)\boldsymbol{\alpha}(I) : \kappa \in \mathbb{S}^2\}$. The reconstruction $\mathbf{X}(\mathbf{y}; I)$ specific to the pixel $\mathbf{y}$ is instead given by composition with the canonical map:

$$\mathbf{X}(\mathbf{y}; I) = B(\kappa)\boldsymbol{\alpha}(I), \quad \text{where} \quad \kappa = \Phi(\mathbf{y}; I). \tag{1}$$

**Viewpoint.** As done in NR-SFM, we assume that the 3D reconstruction is 'viewpoint-free', meaning that the viewpoint is modelled not as part of the deformation parameters $\boldsymbol{\alpha}(I)$, but explicitly, as a separate rigid motion $(R(I), \mathbf{t}(I)) \in \mathbb{SE}(3)$. The rotation $R$ is regressed from the input image in the form proposed by Zhou et al. [75], and translation $\mathbf{t}(I)$ is found by minimizing the reprojection, see Section 3.2 for details. We assume to know the perspective/ortographic *camera model* $\pi : \mathbb{R}^3 \to \mathbb{R}^2$ mapping 3D points in the coordinate frame of the camera to 2D image points (see sup. mat. for details). With this, we can recover the coordinates $\mathbf{y}$ of a pixel from its 3D reconstruction $\mathbf{X}(\mathbf{y}; I)$ as:

$$\mathbf{y} = \pi\left(R(I)\mathbf{X}(\mathbf{y}; I) + \mathbf{t}(I)\right). \tag{2}$$

Note that $\mathbf{y}$ appears on both sides of eq. (2); this lets us define the self-consistency constraint (5).

**Texture.** In addition to the deformation operator $B(\kappa)$, any intrinsic property can be descried in a similar manner. An important example is reconstructing the *albedo* $I(\mathbf{y})$ of the object, which we model as:

$$I(\mathbf{y}) = C(\kappa; \boldsymbol{\beta}(I)), \quad \kappa = \Phi(\mathbf{y}; I), \tag{3}$$

where $C(\kappa; \boldsymbol{\beta})$ maps a small number of image-specific texture parameters $\boldsymbol{\beta}(I) \in \mathbb{R}^{D'}$ to the color of landmark $\kappa$. In Section 4.1, we use this model to transfer texture between images of different objects.

**Implementation via neural networks.** The model above includes several learnable functions that are implemented as deep neural networks. In particular, the canonical map $\Phi(I)$ is implemented

as an image-to-image convolutional network (CNN) with an $\mathbb{R}^{3 \times H \times W}$ input (a color image) and an $\mathbb{R}^{3 \times H \times W}$ output (the spherical embedding). The last layer of this network normalizes each location in $\ell^2$ norm to project 3D vectors to $\mathbb{S}^2$. Functions $\boldsymbol{\alpha}(I)$, $\boldsymbol{\beta}(I)$ and $R(I)$ predicting deformation, texture and viewpoint rotation are also implemented as CNNs. Translation $\mathbf{t}$ is found by minimising the reprojection, as explained below. Finally, functions $B(\kappa)$ and $C(\kappa)$ mapping embeddings to their 3D deformation and texture models are given by multi-layer perceptrons (MLP). The latter effectively allows $\kappa$, and the resulting 3D and texture reconstruction, to have arbitrary resolution.

## 3.2 Learning formulation

The forward pass and loss functions are shown in Figure 2. In order to train C3DM, we assume available a collection of independently-sampled views of an object category $\{I_n\}_{n=1}^N$.[2] Furthermore, for each view, we require annotations for the silhouette $\Omega_n$ of the object as well as the 2D locations of $K$ landmarks $Y_n = (\mathbf{y}_{n1}, \ldots, \mathbf{y}_{nK})$. In practice, this information can often be extracted automatically via a method such as Mask R-CNN [25] and HRNet [54], which we do for most experiments in Section 4. Note that C3DM requires only a small finite set of $K$ landmarks for training, while it learns to produce a continuous landmark map. We use the deformation basis from an NR-SFM method as a prior and add a number of consistency constraints for self-supervision, as discussed next.

**NR-SFM Prior.** Since our model generalizes standard parametric approaches, we can use any such method to bootstrap and accelerate learning. We use the output of the recent C3DPO [44] algorithm $\mathcal{A}_n^* = (B_n^*, \mathcal{V}_n^*, \boldsymbol{\alpha}_n^*, R_n^*)$ in order to anchor the deformation model $B(\kappa)$ in a visible subset $\mathcal{V}_n^*$ of $K$ discrete landmarks, as well as the deformation and viewpoint parameters, for each training image $I_n$.

Note that, contrary to C3DM, C3DPO takes as input the 2D location of the sparse keypoints both at training *and test time*. Furthermore, it can only learn to lift the keypoints for which ground-truth is available at training time. In order to learn C3DM, we thus need to learn from scratch the deformation and viewpoint networks $\boldsymbol{\alpha}(I)$ and $R(I)$, as well as the continuous deformation network $B(\kappa)$. This is necessary so that at test time C3DM can reconstruct the object in a dense manner given only the image $I$, not the keypoints, as input. At training time, we supervise the deformation and viewpoint networks from the C3DPO output via the loss:

$$\mathcal{L}_{\mathrm{pr}}(\Phi, B, \boldsymbol{\alpha}, R; I, Y, \mathcal{A}^*) = \frac{1}{|\mathcal{V}^*|} \sum_{k \in \mathcal{V}^*} \|B(\Phi(\mathbf{y}_k; I)) - B_k^*\|_\epsilon + w_{\boldsymbol{\alpha}} \|\boldsymbol{\alpha}(I) - \boldsymbol{\alpha}^*\|_\epsilon + w_R d_\epsilon(R(I); R^*),$$
(4)

where $\|z\|_\epsilon$ is the pseudo-Huber loss [14] with soft threshold $\epsilon$ and $d_\epsilon$ is a distance between rotations.[3]

**Projection self-consistency loss.** As noted in Section 2, the composition of eqs. (1) and (2) must yield the identity function. This is captured by the *reprojection consistency loss*

$$\mathcal{L}_{\mathrm{repro}}(\Phi, B, \boldsymbol{\alpha}, R; \Omega, I) = \min_{\mathbf{t} \in \mathbb{R}^3} \sum_{\mathbf{y} \in \Omega} \|\hat{\mathbf{y}}(\mathbf{t}) - \mathbf{y}\|_\epsilon, \quad \hat{\mathbf{y}}(\mathbf{t}) = \pi\Big(R(I)\, B(\Phi(\mathbf{y}; I))\, \boldsymbol{\alpha}(I) + \mathbf{t}\Big). \quad (5)$$

It causes the 3D reconstruction of an image pixel $\mathbf{y}$, which is obtained in a viewpoint-free space, to line up with $\mathbf{y}$ once the viewpoint is accounted for. We found optimizing over translation $\mathbf{t}$ in eq. (5) to obtain $\mathbf{t}(I, \Omega, \Phi, B, \boldsymbol{\alpha}, R)$ based on the predicted shape to be more accurate than regressing it directly. Refer to **??** in sup. mat. for optimization algorithm. We use the obtained value as the translation prediction $\mathbf{t}(I)$, in particular, in eq. (6), only implying the dependency on the predictors to simplify the notation. We backpropagate gradients from all losses through this minimization though.

**Apperance loss.** Given two views $I$ and $I'$ of an object, we can use the predicted geometry and viewpoint to establish dense correspondences between them. Namely, given a pixel $\mathbf{y} \in \Omega$ in the first image, we can find the corresponding pixel $\hat{\mathbf{y}}'$ in the second image as:

$$\hat{\mathbf{y}}' = \pi\Big(R(I')\, B(\Phi(\mathbf{y}; I))\, \boldsymbol{\alpha}(I') + \mathbf{t}(I')\Big). \quad (6)$$

This equation is similar to eq. (5), in particular, the canonical map is still computed in the image $I$ to identify the landmark, however the shape $\boldsymbol{\alpha}$ and viewpoint $(R, \mathbf{t})$ are computed from another

image $I'$. Assuming that color constancy holds, we could then simply enforce $I(\mathbf{y}) \approx I'(\hat{\mathbf{y}}')$, but this constraint is violated for non-Lambertian objects or images of different object instances. We thus relax this constraint by using a *perceptual loss* $\mathcal{L}_{\text{percep}}$, which is based on comparing the activations of a pre-trained neural network instead [72]. Please refer to **??** in the sup. mat. for details.

Due to the robustness of the perceptual loss, most images $I$ can be successfully matched to a fairly large set $\mathcal{P}_I = \{I'\}$ of other images, even if they contain a different instance. To further increase robustness to occlusions and illumination differences caused by change of viewpoint, we follow Khot et al. [33]: given a batch of training images $\mathcal{P}_I$, we compare each pixel in $I$ only to $k \leq |\mathcal{P}_I|$ its counterparts in the batch that match the pixel best. This bring us to the following formulation:

$$\mathcal{L}_{\text{percep}}^{\text{min-k}}(\Phi, B, \boldsymbol{\alpha}, R, \mathbf{t}; \Omega, I, \mathcal{P}_I) = \frac{1}{k} \sum_{\mathbf{y} \in \Omega} \min_{Q \subset \mathcal{P}_I : |Q| = k} \sum_{I' \in Q} \mathcal{L}_{\text{percep}}(\Phi, B, \boldsymbol{\alpha}, R, \mathbf{t}; \mathbf{y}, I, I'). \quad (7)$$

**Learning the texture model.** The texture model $(C, \beta)$ can be learned in a similar manner, by minimizing the combination of the photometric and perceptual (7) losses between the generated and original image. Please refer to the supplementary material for specific loss formulations. We do not back-propagate their gradients beyond the appearance model as it deteriorates the geometry.

**Camera-embedding alignment.** We use another constraint that ties the spherical embedding space and camera orientation. It forces the model to use the whole embedding space and avoid re-using its parts for the regions of similar appearance, such as left and right sides of a car. We achieve it by aligning the direction of the mean embedding vector $\kappa$ with the camera direction, minimizing

$$\mathcal{L}_{\text{emb-align}}(\Phi, R; \Omega, I) = \begin{bmatrix} 0 & 0 & 1 \end{bmatrix} R(I) \frac{\bar{\kappa}}{\|\bar{\kappa}\|}, \quad \text{where } \bar{\kappa} = \frac{1}{|\Omega|} \sum_{\mathbf{y} \in \Omega} \Phi(\mathbf{y}; I). \quad (8)$$

**Mask reprojection loss.** We observed that on some datasets like CUB Birds, the reconstructed surface tends to be noisy due to some parts of the embedding space overfitting to specific images. To prevent it interfering with other images, we additionally minimize the following simple loss function:

$$\mathcal{L}_{\text{mask}}(B, \boldsymbol{\alpha}, R, \mathbf{t}; \Omega) = \int_{\mathbb{S}^2} \left[\!\!\left[ \pi \Big( R\, B(\kappa)\, \boldsymbol{\alpha} + \mathbf{t} \Big) \notin \Omega \right]\!\!\right] d\kappa, \quad (9)$$

where we approximate the integration by taking a uniform sample of 1000 points $\kappa$ on a sphere.

## 4 Experiments

We evaluate the proposed method on several datasets using 3D reconstruction metrics. It is difficult to quantitatively evaluate canonical maps, as discussed in Section 4.1. Since reconstruction relies on having a good model for canonical mapping, we use that application to demonstrate the quality of produced maps. For visual evaluation, we use another application, texture transfer, in Figure 3.

**Implementation details.** We build on the open-source implementation of C3DPO for pre-processing[4] and set $\boldsymbol{\alpha} \in \mathbb{R}^{10}$, $\boldsymbol{\beta} \in \mathbb{R}^{128}$. The canonical map network $\Phi$ uses the Hypercolumns architecture [23] on top of ResNet-50 [24], while basis and texture networks $B$ and $C$ are MLPs. See **????** in sup. mat. for description of the architecture, hyperparameters and optimization.

**Benchmarks.** We evaluate the method on a range of challenging datasets. We use C3DM to generate from each test image: (1) a full 360° shape reconstruction as a point cloud $\{B(\kappa)\boldsymbol{\alpha}(I) : \kappa \in \mathcal{K}\}$, where $\mathcal{K}$ consists of 30k sampled embeddings from random training set images, and (2) a depth map from the estimated image viewpoint obtained for each pixel $\mathbf{y} \in \Omega$ as the coordinate $z$ of $R\mathbf{X}(\mathbf{y}; I)$. We compare the full reconstructions against ground-truth point clouds using symmetric Chamfer distance $d_{\text{pcl}}$ (after ICP alignment [9]) and, whenever the dataset has depth maps or calibrations to project the ground-truth meshes, predicted depth maps against ground-truth depth maps as the average per-pixel depth error $d_{\text{depth}}$. In particular, to compute the symmetric Chamfer distance between the predicted and ground-truth point clouds $d_{\text{pcl}}(\hat{C}, C)$, we first correct the scale ambiguity by normalising the variance of the predicted point cloud to match ground truth. Then, we align them

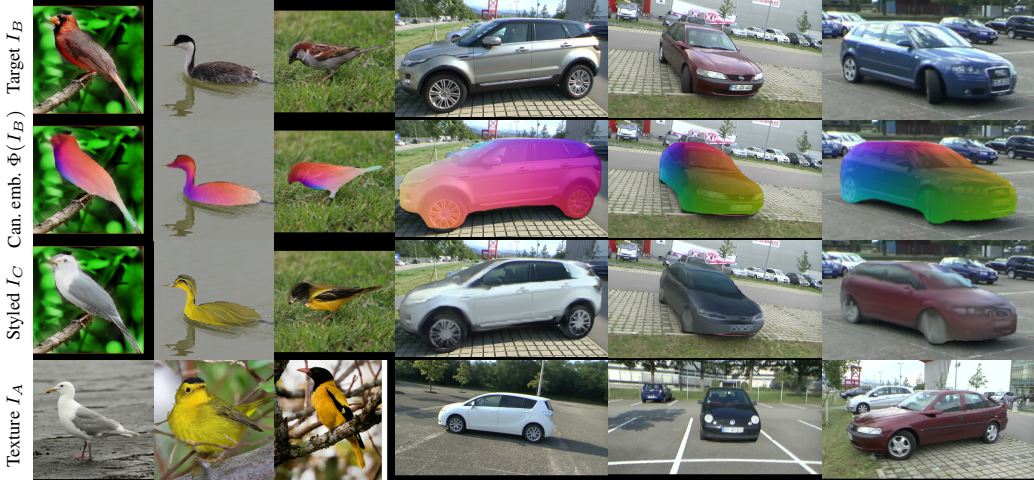

Figure 3: **Canonical mapping and texture transfer for CUB and Freiburg Cars**. Given a target image $I_B$ (1st row), C3DM extracts the canonical embeddings $\kappa = \Phi(\mathbf{y}; I_B)$ (2nd row). Then, given the appearance descriptor $\boldsymbol{\beta}(I_A)$ of a texture image $I_A$ (4th row), the texture network $C$ transfers its style to get a styled image $I_C(\mathbf{y}) = C(\Phi_{\mathbf{y}}(I_B); \boldsymbol{\beta}(I_A))$ (3rd row), which preserves the geometry of the target image $I_B$. Note that we model the texture directly rather than warp the source image, so even the parts occluded in the source image $I_A$ can be styled (5th and 6th columns).

with ICP to obtain the $\tilde{C} = sR\hat{C} + \mathbf{t}$ rigidly aligned with $C$. We define Chamfer distance as the mean $\ell^2$ distance from each point in $C$ to its nearest neighbour in $\tilde{C}$ and make it symmetric:

$$d_{\text{pcl}}(\hat{C}, C) = \frac{1}{2}\Big(d_{Ch}(\tilde{C}, C) + d_{Ch}(C, \tilde{C})\Big), \quad \text{where } d_{Ch}(\tilde{C}, C) = \frac{1}{|C|}\sum_{\mathbf{X} \in C} \min_{\tilde{\mathbf{X}} \in \tilde{C}} \|\tilde{\mathbf{X}} - \mathbf{X}\|. \quad (10)$$

To compute the average per-pixel error between the predicted and ground-truth depth maps $d_{\text{depth}}(\hat{D}, D)$, we first normalize the predicted depth to have the same mean and variance as ground truth within the object mask $\Omega$ in order to deal with the scale ambiguity of 3D reconstruction under perspective projection. Then, we compute the mean absolute difference between the the resulting depth maps within $\Omega$ as $d_{\text{depth}}(\hat{D}, D) = \frac{1}{|\Omega|}\sum_{\mathbf{y} \in \Omega} |\hat{D}_{\mathbf{y}} - D_{\mathbf{y}}|$.

We evaluate on **Freiburg Cars** [52] dataset, containing videos of cars with ground truth SfM/MVS point clouds and depth maps reporting $d_{\text{pcl}}$ and $d_{\text{depth}}$. In order to prove that C3DM can learn from independent views of an object category, we construct training batches so that the appearance loss (6) compares only images of *different* car instances. We further compare our model to the previously published results on a non-rigid category of human faces, training it on **CelebA** [40] and testing it on **Florence 2D/3D Face** [8]. The latter comes with ground-truth point clouds but no depth maps, so we report $d_{\text{pcl}}$ for the central portion of the face. As viewpoints don't vary much in the face data, we also consider **CUB-200-2011 Birds** [64], annotated with 15 semantic 2D keypoints. It lacks 3D annotations, so we adopt the evaluation protocol of CMR [31] and compare against them qualitatively. We compare to CMR using $d_{\text{pcl}}$ on 4 categories from **Pascal3D+** [68], which come with approximate ground-truth shapes obtained by manual CAD model alignment. We trained and ran HRNet [54] to produce input keypoints for evaluation, and also for training where there exists a different dataset to train the detector, i.e. for cars and faces. See **??** for details.

**Baseline.** Our best direct competitor is CMR [31]. For CUB, we use the pre-trained CMR models made available by the authors, and for the other datasets we use their source code to train new models, making sure to use the same train/test splits. For depth evaluation, we convert the mesh output of CMR into a depth map using the camera parameters estimated by CMR, and for shape evaluation, we convert the mesh into a point cloud by uniformly sampling 30k points on the mesh.

### 4.1 Evaluating the canonical map

First, we evaluate the learned canonical map $\Phi_{\mathbf{y}}(I)$ qualitatively by demonstrating that it captures stable object correspondences. In row 2 of Figure 3, we overlay image pixels with color-coded 3D

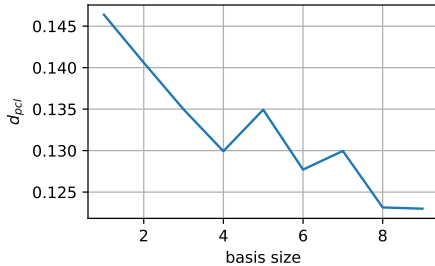

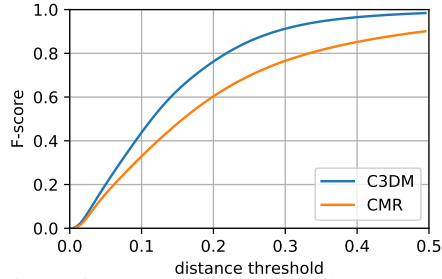

Figure 4: $d_{\mathrm{pcl}}$ as a function of basis and shape descriptor size on Freiburg Cars.

Figure 5: F-score of point cloud reconstruction at variable thresholds on Freiburg Cars.

canonical embedding vectors $\kappa = \Phi_{\mathbf{y}}(I)$. The figure shows that the embeddings are invariant to viewpoint, appearance and deformation. Next, we make use of the texture model (3) to perform texture transfer. Specifically, given a pair of images $(I_A, I_B)$, we generate an image $I_C(\mathbf{y}) = C(\Phi_{\mathbf{y}}(I_B); \boldsymbol{\beta}(I_A))$ that combines the geometry of image $I_B$ and texture of image $I_A$. Row 3 of Figure 3 shows texture transfer results for several pairs of images from our benchmark data.

Previous work used keypoint detection error to evaluate the quality of canonical mapping and shape reconstruction. We argue that it is a biased measure that is easy to satisfy even with degenerate 3D reconstruction or poor canonical mapping outside the keypoints. We evaluated the percentage of correct keypoints (PCK@0.1) as 85%, much higher than CSM [35] (48%) or CMR [31] (47%). This reflects the fact that C3DM is non-parametric and is supervised with keypoint locations through the basis (4) and reprojection (5) losses, so it can easily learn a good keypoint detector. As we see in row 2 of Figure 3 though, canonical maps are not discontinuous, thus not overfit to keypoint locations.

## 4.2 Evaluating 3D reconstructions

**Ablation study.** In Section 4.1, we vary the size of the shape descriptor $\boldsymbol{\alpha}$ and, consequently, the number of blendshapes in $B$. The left-most point corresponds to rigid reconstruction. This sanity check shows that the method can model shape variation rather than predicting generic shape.

In Table 1, we evaluate the quality of 3D reconstruction by C3DM trained with different combinations of loss functions. It shows that each model components improves performance across all metrics and datasets. The contribution of the appearance loss (7) is higher for cars, where the keypoints are sparse; for faces, on the other hand, the network can get far by interpolating between the embeddings of the 98 landmarks even without appearance cues. The camera-embedding alignment loss (8) is also more important for cars because of the higher viewpoint diversity.

The last row in Table 1 evaluates the baseline where we replace our representation with a mesh of a fixed topology, regressing basis vectors at mesh vertices and rasterising their values for image pixels, keeping the same loss functions. CSM [35] uses a similar procedure to define the cycle consistency

| Active Losses $\mathcal{L}$ | | | | Fl. Face | Frei. Cars | |
|---|---|---|---|---|---|---|
| repro | basis | min-k percep | emb-align | $d_{\mathrm{pcl}}$ | $d_{\mathrm{depth}}$ | $d_{\mathrm{pcl}}$ |
| | ✓ | ✓ | ✓ | 6.582 | 0.548 | 0.247 |
| ✓ | | ✓ | ✓ | 7.406 | 0.550 | 0.462 |
| ✓ | ✓ | | ✓ | 5.647 | 0.361 | 0.141 |
| ✓ | ✓ | ✓ | | 5.592 | 0.498 | 0.186 |
| ✓ | ✓ | ✓ | ✓ | **5.574** | **0.311** | **0.123** |
| interpolation thru mesh | | | | 13.721 | 0.596 | 0.182 |

Table 1: **3D reconstruction accuracy for C3DM variants on cars and faces.** We evaluate disabling losses (5), (7), (8), and the first term in (4), one by one.

| Dataset | CMR [31] | C3DM |
|---|---|---|
| Flo. Face | 13.09 | **5.57** |
| Frei. Cars | 0.20/0.50 | **0.12/0.31** |
| P3D Plane | 0.022 | **0.019** |
| P3D Chair | 0.049 | **0.043** |
| P3D Car | 0.028 | 0.028 |
| P3D Bus | 0.037 | **0.036** |

Table 2: $d_{\mathbf{pcl}}$ **on Freiburg Cars, Florence Face, and Pascal 3D+** comparing our method to CMR [31]. For Frei. Cars, $d_{\mathrm{depth}}$ is also reported after slash.

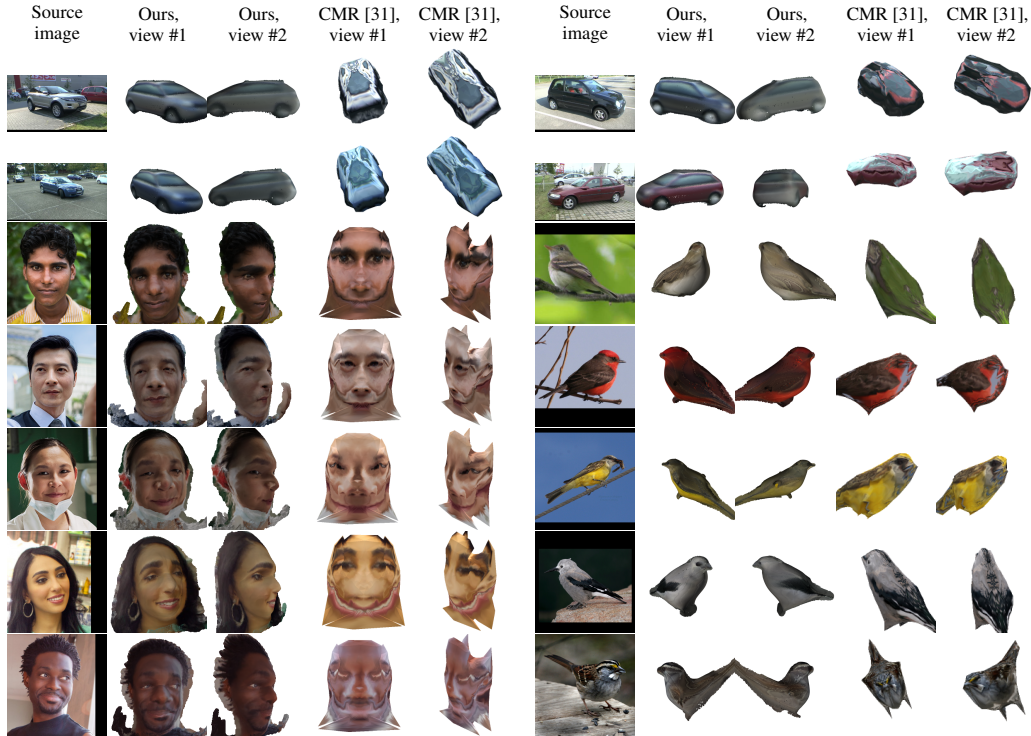

| Source image | Ours, view #1 | Ours, view #2 | CMR [31], view #1 | CMR [31], view #2 | Source image | Ours, view #1 | Ours, view #2 | CMR [31], view #1 | CMR [31], view #2 |

Figure 6: **Visual comparison of the results on Freiburg Cars (top two rows), human faces (left column), and CUB Birds (right column).** For each dataset, we show the source image (1st column), C3DM and CMR reconstructions from the original viewpoint (*view #1*, 2nd and 4th columns, respectively) and from an alternative viewpoint (*view #2*, 3rd and 5th columns).

loss, with the difference that we rasterise basis vectors $B(\kappa)$ rather than 3D coordinates $\mathbf{X}$. It allows us to compute the basis matching loss in eq. (4), which is defined on keypoints that do not have to correspond to mesh vertices. On our data, training does not converge to a reasonable shape, probably because the added layer of indirection through mesh makes backpropagation more difficult.

**Comparison with the state-of-the-art.** Table 2 compares the Chamfer distance $d_{\text{pcl}}$ and depth error $d_{\text{depth}}$ (where applicable) of C3DM against CMR [31]. On Freiburg Cars and Florence Face, our method attains significantly better results than CMR. C3DM produces reasonble reconstructions and generally outperforms CMR on four categories from Pascal3D+ with big lead on chairs. Section 4.1 shows that C3DM attains uniformly higher F-score (as defined by Tatarchenko et al. [56]) than CMR on Frei. Cars. The visualisations in Figure 6 confirm that C3DM is better at modelling fine details.

On Freiburg Cars, our method can handle perspective distortions better and is less dependent on instance segmentation failures since it does not have to satisfy the silhouette reprojection loss. On CelebA, CMR, which relies on this silhouette reprojection loss, produces overly smooth meshes that lack important details like protruding noses. Conversely, C3DM leverages the keypoints lifted by C3DPO to accurately reconstruct noses and chins. On CUB Birds, it is again apparent that C3DM can reconstruct fine details like beaks. See **??** and videos for more visual results.

## 5 Conclusions

We have presented C3DM, a method that learns under weak 2D supervision to densely reconstruct categories of non-rigid objects from single views, establishing dense correspondences between them in the process. We showed that the model can be trained to reconstruct diverse categories such as cars, birds and human faces, obtaining better results than existing reconstruction methods that work under the same assumptions. We also demonstrated the quality of dense correspondences by applying them to transfer textures. The method is still limited by the availability of some 2D supervision (silhouettes and sparse keypoints) at training time. We aim to remove this dependency in future work.

## Potential broader impact

Our work achieves better image-based 3D reconstruction than the existing technology, which is already available to the wider public. While we outperform existing methods on benchmarks, however, the capabilities of our algorithm are not sufficiently different to be likely to open new possibilities for misuse.

Our method interprets images and reconstructs objects in 3D. This is conceivably useful in many applications, from autonomy to virtual and augmented reality. Likewise, it is possible that this technology, as any other, could be misused. However, we do not believe that our method is more prone to misuse than most contributions to machine learning.

As for any research output, there is an area of uncertainty on how our contributions could be incorporated in future research work and the consequent impact of that. We believe that our advances are methodologically significant, and thus we hope to have a positive impact in the community, leading to further developments down the line. However, it is very difficult to predict the nature of all such possible developments.

## Funding disclosure

The authors have not received any third party funding related to this work.

## Acknowledgements

We want to thank Nikhila Ravi for sharing CMR models trained on Pascal3D+ and NeurIPS reviewers for their valuable suggestions.

## Footnotes

[2] "Independent" means that views contain different object deformations or even different object instances.

[3] $\|z\|_\epsilon = \epsilon(\sqrt{1 + (\|z\|/\epsilon)^2} - 1)$; it behaves as a quadratic function of $\|z\|$ in the vicinity of 0 and a linear one when $\|z\| \to \infty$, which makes it both smooth and robust to outliers. See sup. mat. for definition of $d_\epsilon$.

[4]`https://github.com/facebookresearch/c3dpo_nrsfm`

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
