[Supplementary Material]

# Canonical 3D Deformer Maps:
# Unifying parametric and non-parametric methods for dense weakly-supervised category reconstruction

## *Supplementary material*

**David Novotny**[*]    **Roman Shapovalov**[*]    **Andrea Vedaldi**

Facebook AI Research

{dnovotny, romansh, vedaldi}@fb.com

http://www.robots.ox.ac.uk/~david/c3dm/

## A  Architecture details

Figure I shows the backbone of our architecture, together with the basis and texture predictors $B$ and $C$. The trunk of C3DM consists of a Feature Pyramid Network pre-trained on ImageNet. In more detail, Conv-Upsample blocks are attached to the outputs of each of the Res1, Res2, Res3 and Res4 layers of a ResNet50. Each Conv-Upsample outputs a tensor with the spatial resolution of the first auxiliary branch that takes Res1 as an input. The four tensors are then summed and $\ell^2$-normalized in order to produce the canonical embedding tensor $\kappa$.

The insets of Figure I show the architecture of the basis and texture networks $B(\kappa)$ and $C(\kappa, \boldsymbol{\beta}(I))$. The networks follow the C3DPO [10] architecture. Each of them consists of a fully connected (FC) layer, followed by three fully connected residual blocks (shown in detail in the lower-right inset) and another fully connected layer adapting the output dimensionality. The LayerNorm layers [1] used in these networks only perform $\ell^2$ normalization across channels, without using trainable parameters. The basis network takes as input the map of 2D canonical embeddings $\kappa$, while the texture network concatenates them with the same texture descriptor $\boldsymbol{\beta}$ to get the 130-dimensional vector for each pixel. The basis network outputs the 30-dimensional vector for each pixel (10 3-dimensional basis vectors), while the texture network outputs 3D per-pixel colors.

Figure II extends the diagram with the computations specific to the training time. For supervision, the training also runs C3DPO on 2D keypoints and uses the predictions and bases to define the NR-SFM prior loss (4) in the maon paper. The diagram also shows the reprojection consistency loss (5) in the main paper, cross-image perceptual loss (7) in the main paper, which requires the viewpoint and shape predictions for other images in the batch, camera-embedding alignment loss (8) in the main paper, and the texture model loss (3).

**Batch sampling.**  In each training epoch, we sample 3000 batches of 10 random images (adding a constraint on Freiburg Cars that they don't come from the same sequence). We optimize the network using SGD with momentum, starting with learning rate 0.001 and decreasing $10\times$ whenever the objective plateaus. We stop training after 50 epochs.

Since most datasets are biased in terms of the viewpoints, e.g. birds are less likely to be photographed from the front or back than from the side, we apply inverse propensity correction on the distribution of 1D rotations to ensure uniform coverage. We correct the distribution of rotations in the horizontal plane only, assuming that the pitch varies less than the azimuth, which is true for most object-centric datasets. In particular, we first find the upward direction as an eigenvector of the rotation axes extracted from the camera orientations extracted by NR-SFM from the training set: $\{R_n^*\}$.

---

[*]Authors contributed equally.

Figure I: **The detailed architecture of prediction-time C3DM flow.** All networks share the common ResNet50 backbone. Camera orientation, shape and texture parameters are regressed from the final residual layer. The embedding prediction network $\Phi$ processes outputs of the four residual blocks with the Conv-Upsample subnetwork shown in the left inset, then sums and normalises their outputs to obtain the map of spherical embeddings $\kappa$. They are passed through basis and texture networks that share the architecture, which is shown in the middle and right insets. Finally, the predicted basis vectors are multiplied by shape parameters $\boldsymbol{\alpha}$ to obtain 3D reconstruction of the visible points.

Then we compute the azimuth $a(R_n^*)$ as the rotation component around the estimated upward axis. The sampling weight for an image $I_n$ is thus found as $\left(p(a(R_n^*))\right)^{-1}$, where the distribution $p$ is approximated by a histogram of 16 bins. Note that we only need to do this at training time when NR-SFM viewpoint predictions are available; at test time, the networks can take a single image.

To compute the min-k cross-image perceptual loss (7) in the main paper, we treat the first image $I$ in the batch as a target and warp the rest of the images using their *estimated* camera and shape parameters $R(I'), \mathbf{t}(I'), \boldsymbol{\alpha}(I')$. For each pixel, we average the distances to $k = 6$ closest feature maps as per eq. (7).

**Implementation.** We implemented C3DM using Pytorch framework. We run training on a single NVidia Tesla V100 GPU with 16 Gb of memory. Training for full 50 epochs takes around 48 hours.

**Runtime analysis** On a single gpu, the feedforward pass of our network takes one average 0.111 sec per image.

# B Details of the photometric and perceptual losses

To enforce photometric consistency, we can use the following loss:

$$\mathcal{L}_{\text{photo}}(I'; \Omega, I) = \sum_{\mathbf{y} \in \Omega} \|I'(\mathbf{y}) - I(\mathbf{y})\|_\epsilon. \tag{1}$$

Here $I$ and $I'$ are two images, $\Omega$ is the region of image $I$ that contains the object (*i.e.* the object mask).

To capture higher-level consistency between images, in particular in the cross-image consistency loss (6) in the main paper between the target image and warped reference image, we use *perceptual loss* $\mathcal{L}_{\text{percep}}$ that compares the activations of a pre-trained neural network [17]. Specifically, we compute pseudo-Huber loss between the activations of a VGG network, averaged over several layers. The perceptual loss uses the pretrained VGG-19 network [12]. Let $\Psi_l(I)$ be the layer $l$ activations of VGG-19 fed by the image $I$. We then define the perceptual loss as

$$\mathcal{L}_{\text{percep}}(I'; \Omega, I) = \sum_{\mathbf{y} \in \Omega} \sum_{l \in \{0,5,10,15\}} \left\| \text{upsample}\big(\Psi_l(I') - \Psi_l(I)\big)[\mathbf{y}] \right\|_\epsilon, \tag{2}$$

Figure II: **The training time C3DM flow,** where the backbones showed in Figure I are collapsed to the boxes with ellipses. We supervise the predicted basis map with C3DPO bases at keypoint locations. At training time, we also run C3DPO on 2D keypoints to supervise shape parameters and camera orientation. Embedding alignment loss acts on the estimated camera orientation and average spherical embeddings. We project the 3D reconstruction using the estimated camera parameters to define the reprojection consistency loss. To define the cross-image perceptual consistency loss, we run our network on another image (in practice, the other images in the batch are used) and use its shape and camera parameters to project the estimated basis vectors and compare with that image. Finally, we supervise the output of the texture model with the original image.

where upsample() interpolates the feature map to the match the resolution of the network input.

We can now formally define the optimisation problem for the texture model described in Section 3.1. Given the input image $I$ and 2D embeddings for all its pixels $\kappa$, it re-produces the image $I'$ using $I'(\mathbf{y}) = C(\kappa(\mathbf{y}); \boldsymbol{\beta}(I))$. The weights of neural networks implementing $C$ and $\boldsymbol{\beta}$ are found by minimising

$$\mathcal{L}_{\text{tex}}(I'; \Omega, I) = w_{\text{photo}}\mathcal{L}_{\text{photo}}^{\text{tex}}(I'; \Omega, I) + w_{\text{percep}}\mathcal{L}_{\text{percep}}^{\text{tex}}(I'; \Omega, I). \tag{3}$$

Please note again that the gradients of $\mathcal{L}_{\text{tex}}$ are not propagated beyond $\kappa$ to preserve its sole dependence on geometry.

## C  Camera models and ray-projection loss

**Camera models.**    We have to define a camera model $\pi : \mathbb{R}^3 \rightarrow \mathbb{R}^2$ mapping 3D points in the coordinate frame of the camera to 2D image points in order to compute reprojection and photometric losses. If the camera calibration is unknown (as in CelebA, Florence Face, CUB, Pascal 3D+ datasets), we use an *orthographic camera* $\pi(\mathbf{X}) = [x_1, x_2]^{\top}$ where $\mathbf{X} = [x_1, x_2, x_3]^{\top}$. In this case, we also set $\mathbf{t} = 0$ as translation can be removed by centering the 2D data [10] in pre-processing.

If the camera calibration is known (in Freiburg Cars), we can also use a more accurate *perspective camera model* instead:

$$\pi(\mathbf{X}) = \frac{f}{x_3}\begin{bmatrix} x_1 \\ x_2 \end{bmatrix}, \tag{4}$$

where $f$ is the focal length.

Further to Section 3.1, here, we describe additional implementation details that were important for the success of the perspective projection model on the Freiburg Cars dataset.

**Ray-projection loss**    For perspective model, we have also found an improvement that significantly stabilizes the C3DPO algorithm that we use to constrain C3DM. The idea is to modify reprojection loss to measure, instead of the distance between 2D projections $\mathbf{y}$ and $\hat{\mathbf{y}}$, the distance of the 3D point

$\mathbf{X}(\mathbf{y})$ to the line passing through $\mathbf{y}$ and the camera center. The advantage is removing the division embedded in the perspective projection equation (4).

In order to minimize the reprojection error (5) in the main paper under the perspective projection model, a naïve implementation would minimize the following perspective re-projection loss:

$$\mathcal{L}_{\text{repro}}^{\text{persp}}(\Phi;\Omega,I) = \sum_{\mathbf{y}\in\Omega}\left\|\pi_{I,\mathbf{0}}(\mathbf{X}_{R,\mathbf{t}}(\mathbf{y}))-\mathbf{y}\right\|_{\epsilon}, \tag{5}$$

where $\mathbf{X}_{R,\mathbf{t}}(\mathbf{y}) = R\mathbf{X}(\mathbf{y})+\mathbf{t}$ is the 3D point extracted from pixel $\mathbf{y}$ and expressed in the coordinate frame of the camera of the image $I_n$. Unfortunately, we found that the division in the perspective projection formula $\pi_{I,\mathbf{0}} = \frac{f}{x_3}[x_1\ x_2]^{\top}$ leads to unstable training. This is due to exploding gradient magnitudes caused by 3D points $\mathbf{X}$ predicted to lie too close to the camera projection plane. While this could be extenuated by clamping the points to lie in a safe distance from the camera plane, due to the non-linearity of the projection gradient, the re-projection loss (5) still would not converge stably.

In order to remove the gradient non-linearity, we alter the re-projection loss to the *ray-projection* loss:

$$\mathcal{L}_{\text{repro}}^{\text{ray}}(\Phi;\Omega,I) = \sum_{\mathbf{y}\in\Omega}\left\|\mathbf{X}_{R,\mathbf{t}}(\mathbf{y})-\left[\mathbf{r}(\mathbf{y})^{\top}\mathbf{X}_{R,\mathbf{t}}(\mathbf{y})\right]\mathbf{r}(\mathbf{y})\right\|_{\epsilon}, \tag{6}$$

where $\mathbf{r}(\mathbf{y})$ stands for the direction vector of the projection ray passing through the pixel $\mathbf{y}$ in the image $I$:

$$\mathbf{r}(\mathbf{y}) = \frac{K^{-1}[y_1\ y_2\ 1]^{\top}}{\|K^{-1}[y_1\ y_2\ 1]^{\top}\|},$$

where $K$ is the instrinsic camera calibration matrix. Intuitively, eq. (6) minimizes the orthogonal distance between the the estimated point $\mathbf{X}_{R,\mathbf{t}}(\mathbf{y})$ and its projection on the ground truth projection ray $\mathbf{r}(\mathbf{y})$. We notice that eq. (6) is linear in $\mathbf{X}_{R,\mathbf{t}}$ on infinity and quadratic in the compact region around the optimum, hence the magnitude of the gradient is bounded from above. We found this addition important for convergence of C3DM.

**Perspective projection for C3DPO**    In order to optimize eq. (6), a C3DPO model [10] trained using the perspective projection model is required. Since the original C3DPO codebase only admits orthographic cameras, we will describe additions to the pipeline that enable training a perspective model on Freiburg Cars.

C3DPO optimizes a combination of canonicalization and reprojection losses. To this end, we replace the original C3DPO reprojection loss (eq. (4) in [10]) with the ray-projection loss (6). Additionally, unlike in the orthographic case, one has to determine the full 3DoF position of the camera w.r.t. the object coordinate frame. While it is possible to let C3DPO predict translation as an additional output of the network, we avoid over-parametrization of the problem by estimating camera translation as a solution to a simple least-squares problem.

In more detail, we exploit the locally quadratic form of the ray-projection loss and formulate the translation estimation problem that allows for a closed-form solution. Assuming that C3DPO, given a list of input 2D landmarks $\mathbf{y}_1, ..., \mathbf{y}_K$, predicts a camera rotation matrix $R$, the translation can be obtained as a solution to the following problem:

$$\mathbf{t}^{*} = \text{argmin}_{\mathbf{t}} \sum_{i=1}^{K}\left\|\mathbf{X}_{R,\mathbf{t}}(\mathbf{y}_k)-\mathbf{r}(\mathbf{y}_k)^{\top}\mathbf{X}_{R,\mathbf{t}}(\mathbf{y}_k)\mathbf{r}(\mathbf{y}_k)\right\|^{2}.$$

After a few mathematical manipulations, we arrive at the following closed-form expression for $\mathbf{t}^{*}$:

$$\mathbf{t}^{*} = \left[\sum_{k=1}^{K}(I-\Gamma_k)\right]^{-1}\left[\sum_{k=1}^{K}(\Gamma_k-I)\mathbf{X}_{R,\mathbf{0}}\right], \tag{7}$$

where $\Gamma_k = \mathbf{r}(\mathbf{y}_k)\mathbf{r}(\mathbf{y}_k)^{\top}$ is an outer product of $\mathbf{r}(\mathbf{y}_k)$ with itself. Using eq. (7), we can estimate the camera translation online during the SGD iterations of the C3DPO optimization. Note that the matrix inverse in eq. (7) is not an issue because of the small size of the matrix being inverted ($3\times3$) and the possibility to backpropagate through matrix inversion using modern automatic differentiation frameworks (PyTorch).

## D   Rotation loss

We use the distance between rotation matrices $d_\epsilon(R, R^*)$ as part of the loss (4) in the main paper. We aim to penalise large angular distance, while avoiding the exploding gradients of inverse trigonometric functions. First, we note that the relative rotation can be computed as $R^\top R^*$. Next, converting it to the axis-angle representation lets us compute the angular component as $\theta = \arccos\left(\frac{1}{2}(\mathrm{Tr}(R^\top R^*) - 1)\right)$. Using the fact that $\arccos$ is monotonically decreasing, we strip it and apply an affine transform to make sure the loss achieves the minimum at 0:

$$d_\epsilon(R, R^*) = 1 - \cos\theta = \frac{3 - \mathrm{Tr}(R^\top R^*)}{2}. \tag{8}$$

## E   Datasets

**Freiburg Cars (FrC).**   In order to test our algorithm in a low-noise setting, we consider the Freiburg cars dataset [11][2] containing walkaround videos of 52 cars. While this dataset contains videos of the cars, in order to test the ability of the photometric loss (7) in the main paper to reconstruct objects even if the views are independent, we pair each pivot image $I$ with a selection of other images $\mathcal{P}_I$ extracted from *different* video sequences.

Following Novotny et al. [9, 8], we set out 5 sequences for validation (indexed 22, 34, 36, 37, 42). The training set contains 11,162 training frames and 1,427 validation frames. For evaluation, we also use their ground-truth 3D point clouds, but we only retain the 3D points that, after being projected into each image of a given test sequence, fall within the corresponding segmentation mask. Each point cloud is further normalized to zero-mean and unit variance along the 3 coordinate axes. Please refer to [9] for details.

As an input to our method, we use the pre-trained Mask R-CNN of [4] to extract the segmentation masks and the HRNet [6] trained on PASCAL 3D+ [16] to extract the 2D keypoints. Hence, all inputs to our method are extracted automatically. We excluded the frames where a car was detected with a confidence below a threshold.

We report the Chamfer distance $d_{\mathrm{pcl}}$ between the ground truth and the predicted point clouds after rigid alignment via ICP [3]. The point cloud predictions are obtained as explained in the *Benchmarks* section of the main text, with $|\mathcal{B}| = 30\mathrm{k}$. Furthermore, we evaluate the quality of our depth predictions by measuring the average depth distance $d_{\mathrm{depth}}$ between the point cloud formed by un-projecting the predicted depth map and the visible part of the ground truth point cloud.

**CelebA and Florence faces (FF).**   The FrC dataset contains deformation between object instances, but each object itself is rigid. In order to compare the ability of our method to handle instance-level non-rigid deformations with the CMR's, we also run the method on images of human faces; in particular, we train our algorithm on the training set of CelebA dataset [7][3] containing 161,934 face images and test it on the Florence 2D/3D Face dataset [2][4]. The latter contains videos of 53 people and their ground truth 3D meshes, which we can use to assess the quality of our 3D reconstructions. Following a standard practice, we crop each 3D mesh to retain points that lie within 100mm distance from the nose tip. We extract 98 semantic keypoints for each training and test face using the pre-trained HRNet detector of [13].

For evaluation on FF, five frames are uniformly sampled from each test sequence. We then use our network to reconstruct each test face in 3D and evaluate $d_{\mathrm{pcl}}$ after ICP alignment. Since the extent of the predicted face differs from the ground truth, we first pre-align the prediction by registering a 3D crop that covers the convex hull of the 98 semantic keypoints. The 100mm nose-tip crop is then extracted from the pre-aligned mesh and is aligned for the second time. $d_{\mathrm{depth}}$ is not reported for FF since the dataset does not contain ground truth per-frame depth.

**CUB-200-2011 Birds.** We evaluate our method qualitatively on the CUB Birds dataset [14][5], which consists of 11,788 still images of birds belonging to 200 species. Each image is annotated with 15 semantic keypoints. As done in [10], for evaluation we use detections of a pre-trained HRNet. The dataset is challenging mainly due to significant shape variations across bird species, in addition to instance-level articulation. Since there is no 3D ground truth for that dataset, we qualitatively compare the quality of 3D reconstruction to the ones of CMR [5]. We also use the same training/validation split as CMR.

**Pascal3D+.** We provide additional comparison to CMR on four categories of Pascal3D+ [15][6]: aeroplane, consisting of 1194 training and 1135 test images, bus (674 training / 657 test), car (2765 training / 2713 test), and chair (650 training / 666 test). It has been manually annotated by rigidly aligning one of category-specific CAD models, so the annotation has noisy and biased shape and pose. Since the original CMR codebase contains models for only two classes, we trained CMR models on all considered classes ourselves (using their codebase) and test on the corresponding validation sets. We report only $d_{\text{pcl}}$, since the depth maps obtained by projecting with noisy cameras are unreliable.

## F   Hyperparameters used in experiments

To sum up, during training, we optimize the following weighted sum of loss functions:

$$
\begin{aligned}
\mathcal{L}(\Phi, B, \boldsymbol{\alpha}, R, \mathbf{t}, \hat{I}; \Omega, I, \mathcal{P}_I, \mathcal{A}^*) = \ & w_{\text{pr}}\mathcal{L}_{\text{pr}}(\Phi, B, \boldsymbol{\alpha}, R; I, Y, \mathcal{A}^*) + \\
& w_{\text{repro}}\mathcal{L}_{\text{repro}}(\Phi, B, \boldsymbol{\alpha}, R; \Omega, I) + \\
& w_{\text{percep}}^{\text{min-k}}\mathcal{L}_{\text{percep}}^{\text{min-k}}(\Phi, B, \boldsymbol{\alpha}, R, \mathbf{t}; \Omega, I, \mathcal{P}_I) + \\
& w_{\text{emb-align}}\mathcal{L}_{\text{emb-align}}(\Phi, R; \Omega, I) + \\
& w_{\text{mask}}\mathcal{L}_{\text{mask}}(B, \boldsymbol{\alpha}, R, \mathbf{t}; \Omega) + \\
& \mathcal{L}_{\text{tex}}(\hat{I}; \Omega, I).
\end{aligned}
\tag{9}
$$

We set most weights such that the corresponding term has a magnitude of about 1 in the beginning of training. We set $w_{\text{pr}} = 1$, $w_{\boldsymbol{\alpha}} = 1$, $w_{\text{repro}} = 1$ for the perspective camera model and $w_{\text{repro}} = 0.01$ for the orthographic one, where the error is measured in pixels rather than world units. For the components of texture loss, we set $w_{\text{photo}}^{\text{tex}} = 1$, and $w_{\text{percep}}^{\text{tex}} = 0.1$. Likewise, we set the weight for the geometry perceptual loss $w_{\text{percep}}^{\text{min-k}} = 0.1$. We ran grid search for the camera-related parameters within the following ranges: $w_R \in \{1, 10\}$, and $w_{\text{emb-align}} \in \{1, 10\}$. We enable $\mathcal{L}_{\text{mask}}$ for CUB Birds, Faces, and Pascal3D+ aeroplanes and chairs with weight $w_{\text{mask}} = 1$.

## G   Additional qualitative results

Figures III and IV contain additional single-view reconstruction results. We can see that C3DM is robust to occlusions and instance segmentation failures: the 3D shape is reasonably completed in those cases. Furthermore, Figures V and VI have been populated with supplemental texture transfer results. Note that all images are taken from the test set, and images from the same FrC sequence do not co-occur in training and test sets. We also invite the readers to watch the attached videos of the rendered reconstructions to better evaluate 3D reconstruction quality.

Figure III: **Additional single-view reconstruction results on images from the test sequences of CUB Birds**. Columns: input image; canonical mapping; 3D reconstruction with the reconstructed texture from two viewpoints.

Figure IV: **Additional single-view reconstruction results on images from the test sequences of Freiburg Cars**. Columns: input image; canonical mapping; 3D reconstruction with the reconstructed texture from two viewpoints.

Figure V: **Canonical mapping and texture transfer for CUB.** Given a target image $I_B$ (1st row), C3DM extracts the canonical embeddings $\kappa = \Phi(\mathbf{y}; I_B)$ (2nd row). Then, given the appearance descriptor $\boldsymbol{\beta}(I_A)$ of a texture image $I_A$ (4th row), the texture network $C$ transfers its style to get a styled image $I_C(\mathbf{y}) = C(\Phi_{\mathbf{y}}(I_B); \boldsymbol{\beta}(I_A))$ (3rd row), which preserves the geometry of the target image $I_B$.

Target image $I_B$     Can. emb. $\Phi(I_B)$     Styled image $I_C$     Texture image $I_A$

Figure VI: **Canonical mapping and texture transfer for Freiburg cars.** Given a target image $I_B$ (1$^{st}$ row), C3DM extracts the canonical embeddings $\kappa = \Phi(\mathbf{y}; I_B)$ (2$^{nd}$ row). Then, given the appearance descriptor $\boldsymbol{\beta}(I_A)$ of a texture image $I_A$ (4$^{th}$ row), the texture network $C$ transfers its style to get a styled image $I_C(\mathbf{y}) = C(\Phi_{\mathbf{y}}(I_B); \boldsymbol{\beta}(I_A))$ (3$^{rd}$ row), which preserves the geometry of the target image $I_B$.

## Footnotes

[2]`https://github.com/lmb-freiburg/unsup-car-dataset`

[3]`http://mmlab.ie.cuhk.edu.hk/projects/CelebA.html`

[4]`http://www.micc.unifi.it/masi/research/ffd/` ©Copyright 2011–2019 MICC — Media Integration and Communication Center, University of Florence. The Florence 2D/3D Face Dataset.

[5]http://www.vision.caltech.edu/visipedia/CUB-200-2011.html

[6]https://cvgl.stanford.edu/projects/pascal3d.html