[Reviews · NeurIPS 2020]

Review 1

Summary and Contributions: This paper addresses the problem of inferring shape and texture given a collection of images where each image depicts a different instance of a shape category with known object masks and sparse keypoints during training. This paper incorporates a lot of different pieces, but the main technical contribution is to combine predicting (spherical) surface canonical coordinates with the C3DPO nonrigid SfM pipeline. The paper evaluates on several datasets and quantitatively compares against CMR [29] for shape reconstruction and qualitatively for shape texture transfer.

Strengths: This paper addresses a hard problem. While the main components were demonstrated in previous work (image => canonical coordinates with cycle loss for mapping to a single canonical shape [34], C3DPO for nonrigid shape reconstruction with sparse keypoints [43]), their combination for dense shape reconstruction and texture transfer makes it interesting and timely. The paper also incorporates other technical details for improving the results - appearance loss (used in prior work [32]), soft occlusion regularization, and deformation model for generating texture. The quantitative and qualitative improvement over CMR [29] (state of the art) for shape reconstruction is compelling and model ablations with respect to shape reconstruction look reasonable.

Weaknesses: Suggestions for improvement: 1. The main weakness for me is that the contribution with respect to texture transfer is not fully evaluated. While the textures look qualitatively better than CMR’s in Figure 4, it’s not clear if the difference is solely due to better shape reconstruction. Missing is a comparison where CMR’s texture transfer method is applied to this paper’s proposed shape reconstruction pipeline. Also, it would be great if there is a quantitative evaluation with respect to texture transfer. A few potential ideas come to mind for quantitative evaluation - run a user study, evaluate a pre-trained classifier on the rendered shape with texture, compute FID score. 2. It would be good to see an experiment on how well the proposed method works on keypoint transfer (as is evaluated in the CSM paper [34]) to see whether the C3DPO shape model helps with this task. 3. Both this method and CMR uses ground truth masks and keypoints during training. It would be great to see how well the method performs when both the masks and keypoints are computed automatically during training and test (L147-148). 4. It would be good to have a more detailed discussion of the differences with respect to the shape and texture model of CMR (either in the related work or in Section 3). 5. While this paper just appeared at CVPR 2020, it may be worth mentioning anyways in the related work: Articulation-aware Canonical Surface Mapping. Nilesh Kulkarni, Abhinav Gupta, David Fouhey, Shubham Tulsiani. CVPR, 2020. 6. For B(k) and C(k), it may be good to try a sinusoidal positional encoding, similar to transformers and nerf. 7. It would be good to see F-score reported for shape reconstruction (see the paper "What Do Single-view 3D Reconstruction Networks Learn?” for setting up F-score). In light of these suggestions, I’m still leaning positive on this paper. However, for me, points (1) and (2) are the main limitations with the current paper draft.

Correctness: I did not find any correctness issues.

Clarity: The paper is well written and was a pleasure to read. Some small comments: L113 “most surfaces” => maybe be more precise and say that the shapes are assumed to be genus-0. Section 3 notation: Please be careful with the notation and variable overloading. For example, L115 X(k) and Eq (1) X(y). Equation (2), maybe call left-most “y” as “y_hat” (as is done later). L130 "I(y)”. L204 “where K consists of…” - This sentence was not clear to me. Why are random samples from training images returned for a test image? There are small typos throughout (e.g., L3 “an novel”) - please proofread.

Relation to Prior Work: The references are good.

Reproducibility: Yes

Additional Feedback: Final feedback: After a very healthy discussion after considering the rebuttal and other reviews, I'm still positive on this paper. Please see the meta review for a summary of the key exchanges during the discussion and a list of highly recommended requested changes.


Review 2

Summary and Contributions: This paper tackles the problem of single view 3D reconstrution and learning to do so from image collections with 2D annotations, the same setup as CMR. This paper combines the recent CSM objective and implicit shape and texture basis (similar to AtlasNet-sphere) instead of an explicit shape representation such as meshes used in CMR. Simply put this paper is CMR + CSM initialized with C3DPO, with implicit shape basis instead of a mesh. The result qualitatively looks nice and improves upon CMR quantitatively. However it's unclear if the improvement comes from the change in shape representation, a key ablative experiment is missing, without which is challenging for the community to decipher what the lessons are.

Strengths: - Good quantiative metric on faces and cars - Results look nice, especially that of texture transfer.

Weaknesses: 1. How much of the improvement is coming from the implicit shape representation over meshes? The proposed approach of combining local and global information via CSM's consistency loss could have been done with meshes (CSM was mesh based). What does the result look like with meshes? Or CMR could have also used this implicit shape representation. This key ablative study is missing. 2. The face results on Figure 4, the shortcomings of CMR mainly seems to lie in the mesh representation. How was the CMR initialized on the half-sphere nature of the faces? this should be discussed. 3. The paper says to adopt the evaluation protocol of CMR and compare qualitatively, however there is quantiatvie evaluation on CUB dataset in CMR that can be done via the mask IOU and PCK metric. 4. The paper claims to have a significant gain over CMR on Pascal3D Chairs, however, none of the results from Plane, Char, and Bus are shown in the paper nor the supplemental. This is rather questionable. It's not surprising that the spherical nature of the representation does not work on Chairs. I suspect much of the difference comes in the representation (as implicit sphere is easier to deform than expicit mesh), again calling for the need to evaluate this proposed approach using the same representation. As the proposed idea to combine CMR and CSM can be done on either representations. 5. Also, there are rather adhoc losses such as the min-k perceptive loss and the L_emb-align, which was not used in CMR. 6. Limitation of the proposed approach is not discussed. 7. The paper should cite Groueix et al. AtlasNet: A Papier-Mâché Approach to Learning 3D Surface Generation CVPR 2018, as the implicit 3D surface representation is very similar to Atlasnet-sphere. 8. The need for explicit basis is not clearly motivated -- the ablation seems to indicate however in principal can't the non-linear MLP basis learn this all? Further more it would be interesting to visualize the space of dense basis learned after this process. Unclear points: a. At test time, is silhouette used to visualize the results on Figure 3? What does the cannical map output in the background? This should be clearer. b. Line 193. Why does CUB have this problem of "reconstructed surface tends to be noisy due to some parts .. overfitting to specific images". What is exactly meant by this? And why does the mask reprojection lsos help with this? I don't see the connection between the issue of surface noise and the silhouette loss, beside this loss was also used in CMR. c. Why is line 189 section called soft occlusion? The actual name of the loss is emb-align. The section name is not very intuitive.

Correctness: Yes

Clarity: - Much of the core detail of the paper is in the supplemental. The writing in intro and related work (for ex. sparse NR-SFM section) could be reduced to dedicate more content to the paper. - For one, the ablation study is not well explained due to lack of space. what exactly is "basis" in table 1? The papre lists the equations, 5, 7, 8, 4 but they don't correspond in the same order in the table which is repro (5), basis ?, min-k percep (7), emb-align (8). So basis must be first term in (4) one-by-one, but it's not clear what this means and this should b ediscussed in the papre, not just in the table cpation. In particular this is an important ablation as it's not clear to me why you need the explicit basis when the spherical MLP mapping could in fact capture the deformation as well. - Paragraph from line 115 and in Figure 1 -- at first it was not clear why B has to be a function of k. Why the expression needs to be written as $B(k)\alpha$, instead of $B\alpha(k)$ (take the index after linear combination vs before). This I believe is due to B being implicit, if the representation was explicit, this indexing could have happened after taking the linear combination. I think this should be explained more. - Line 204. k is just a point on the sphere, why not just sample from the sphere, instead of sampling from output of the training? - The sentence in Line 174 is confusing, as in this single-view collection setup there are no two views of the same object? Later in the paragraph it says that the images are of different instances. - Paragraph in line 163 should cite CSM and attribute in this paragraph. - Which 2D loss is required (silhouettes and ladmarks) should be made explicit in the introduction. - Notation of beta around ine 131 is confusing, since B is used for the deformation basis. I think \tau would be better. - In the intro its unclear what is means by intrinsic quantity, giving more concrete example would help.

Relation to Prior Work: The paper could mention atlasnet-sphere, and the exact difference with CMR in related work.

Reproducibility: Yes

Additional Feedback: The results look nice, however, the paper should ablate the improvements that come from the implicit representation and provide the ablation as discussed above. Also the method seems more complicated with lots of losses that weren't used in CMR even though the setup is similar. The approach combines several existing works together: CMR, C3DPO, CSM consistency loss with implicit surface representation. For a publication a better ablative study is necessary and significant improvements in writing. ==================== Post-rebuttal feedback: Thanks for addressing some of my concerns. After an extensive discussion + rebuttal, I am increasing my rating on this paper given that showing how local and global 3D information may be combined is an interesting, very reasonable direction. However, there still remains key experiments and concerns that if addressed would make this paper stronger. We have written these hard requirements out if the paper is to be accepted. Given this I am raising my score to 6: marginally above. ====================


Review 3

Summary and Contributions: This paper proposes Canonical 3D Deformer Map, a representation of category-specific shapes from multi-view images. This representation provides corresponding across different category instances by "anchoring" different objects in a common space. The approach combines a parametric and another non-parameteric representations into canonical maps. Specifically, the parametric representation is provided by non-rigid SfM, and the non-parametric representation comes from the depth prediction. The advantage of the proposed approach is that it not only provides correspondence across different objects, but also dense description of the shape.

Strengths: The method is able to provide shape estimations of non-rigid shapes, which are known to be notoriously hard to reconstruct. In addition, such shape estimations provide correspondence across different objects within the category of interest, as in the original CMR work. The paper is well-written, specifying each loss used and justifying their use by ablation studies. The comparison against CMR shows the model outperforms CMR.

Weaknesses: I am mostly concerned on the contribution of this paper. Building upon CMR, this work incorporates additional non-rigid SfM cues, but it remains unclear to me how this is crucial. Intuitively, this should be helpful when the input images are of the same bird/car/person that is deforming non-rigidly. However, from what the authors present, I cannot figure out how this non-rigid addition is helpful, since what non-rigid motion is present and how it makes CMR fail are unclear. Although the losses are elaborated clearly, but a combination of so many losses is a bit alarming to me as to how well this method actually works. This is not a major concern, though, provided that the author can use better non-rigid input to demonstrate how CMR fails in such cases, and how this work succeeds. The handling of view-dependent effects by relaxing the loss to be perceptual is unsatisfying. To properly handle such effects, the authors may consider modeling viewing directions explicitly, since the camera parameters have been estimated already anyways. Finally, the reliance on non-rigid SfM as a preprocessing step makes me wonder what the subsequent model will then do. This fact that subsequent network training relies on successful non-rigid SfM seems like a major limitation. ==================== Post-rebuttal feedback: Thanks for clarifying on some of my concerns. After extensive discussion, we settled down on a few hard requirements for the paper to be accepted (please see the metareview). I'm not raising my score here as it's hard for me to predict if those requirements could be met. If they were, my score could be regarded as a 6: marginally above. ====================

Correctness: Yes.

Clarity: Mostly yes, but I still don't get why depth prediction by a network is considered non-parametric. I would call that parametric.

Relation to Prior Work: Looks mostly complete.

Reproducibility: Yes

Additional Feedback:


Review 4

Summary and Contributions: This paper extends three existing techniques for image-based shape reconstruction: canonical surface mapping, non-rigid structure from motion, and depth prediction. Instead of predicting depth, it predicts a mapping to a canonical domain (i.e., spherical parameterization of a deformable shape). This canonical mapping, provides corresponding 3D point on all deformable shapes, which are further used in 3D reconstruction with known map to the image. This work nicely extends the current family of research projects on canonical mapping in surface reconstruction. It also leverages various consistency terms and regularizations that are unique to the particular set of representations used in this paper.

Strengths: This paper builds on several state-of-the-art techniques to deliver a robust technique for surface reconstruction with weak supervision. It shows compelling empirical results, and interesting high-level contribution: combining parametric and non-parametric models.

Weaknesses: The novelty of this work is medium (i.e., above NeurIPS bar, but maybe not ground-breaking). It mostly builds on existing well-known loss functions and representations, however, the combination it provides is technically sound, and it is an improvement over prior work.

Correctness: Yes

Clarity: Yes

Relation to Prior Work: Yes

Reproducibility: Yes

Additional Feedback:

[Author Response · NeurIPS 2020]

**Why C3DM is more than CMR/CSM without a mesh.**   Our C3DM representation is *not* a mere drop-in replacement
for the meshes in CMR/CSM. C3DM has major advantages: beyond removing the complexity of differentiable rendering
and re-projecting to a mesh, a key one is that C3DM losses leverage appearance cues (RBG values) to learn the 3D
geometry, while CMR/CSM do not. This may look surprising given that CMR does extract a texture model from the
RGB values, but only silhouette and keypoint supervision affect the geometry (see top of page 9 in [29]). Attempting to
jointly learn generative models for 3D shape and texture is a recipe for failure because such combination has too many
DoFs. Because C3DM generalizes unsupervised monocular depth estimation, we can instead borrow re-projection
losses (e.g. min-k) and use correspondences to constrain the geometry *regardless* of the texture model's quality. Note
that without those appearance cues (in addition to keypoints), CMR fails to reconstruct faces, which C3DM masters.

**Reviewer 1.**   **Texture transfer not evaluated. Evaluate on keypoint transfer.**   Our main contribution is improving
*3D reconstruction* of object categories via a new canonical representation of shape. We use texture transfer as means
to demonstrate the consistency of this canonical map across instances. As suggested, we will also report keypoint
transfer; on CUB, we improve PCK@0.1 drastically: 0.85 vs. 0.48 (CSM) and 0.47 (CMR). Note though that we
use keypoint annotations during training, so the canonical map quality is expected to be better at keypoint locations
than between them. **Cite Kulkarni et al.** OK. **Train and test with automatically detected keypoints.** We *do* use

automatically detected masks/keypoints for training/testing in *all* cases where possible: Freiburg
Cars and FlorenceFace (Appendix E). For P3D and CUB birds, there is no other dataset to
train the keypoint detector, so we use GT annotations for training. **Try sinusoidal embedding**
**for $B, C$.** Thank you; we are planning to experiment with spherical harmonics in the future.
**Report F-score.** We will add the plots to the final version. Results on FreiCars are in the figure
to the right. Consistent with Chamfer distance, C3DM outperforms CMR on all thresholds.

**Reviewer 2.**   **How does [your] method improve over meshes?**   Our representation is not a mere drop-in replacement
for CMR's meshes. Specifically, C3DM innovatively bypasses the complexities of CMR/CSM. It provides a better
performing alternative to the widespread mesh rendering paradigm. Said that, we can indeed convert our representation
to a mesh by warping an icosphere vertices with eq. (1). When done after training, on FreiCars, it increases $d_{pcl}$ from
0.13 to 0.18 due to finite mesh resolution. If used as a representation during training, swapping $\mathcal{L}_{repro}$ and $\mathcal{L}_{percep}^{min-k}$ with
CSM's cycle consistency loss through the mesh further increases $d_{pcl}$ to 0.31, even worse than C3DM without $\mathcal{L}_{repro}$!
We conclude that enforcing cycle conciscency through mesh is not adequate for our setting. **CMR fails on faces. How**
**was it initialized?** For fairness, we did not apply any dataset-specific initialization to any of the benchmarked methods.
CMR fails on faces because it relies on silhouette loss, which is insufficient for learning detailed facial geometry.
**Evaluate mask IOU and PCK metric?** Please refer to the answer to R1 for PCK on CUB. Note that the IOU/PCK
metrics are 2D and do not evaluate 3D reconstruction, e.g. flat 3D shapes with matching deformation/viewpoint can
satisfy them. CMR has to use IOU/PCK because CUB lacks 3D annotations. Our evaluation on the datasets with 3D
ground truth (Freiburg Cars, Florence Face) is thus an improvement over CMR's evaluation on CUB. **Similar to**
**Atlasnet-sphere.** Will cite; indeed, C3DM canonical map is similar to Atlasnet-sphere, but, crucially, the rest of the
pipeline, including handling 3D deformations, focus on real image data and weak supervision, are *significantly different*.
**The explicit basis is not clearly motivated.** We believe that our continuous extension of the sparse NRSfM basis is
novel and appropriately motivates the explicit basis. Other works, including CMR, only re-use the camera parameters
from [NR]SfM, while we also exploit the deformation basis. **Adhoc losses: min-k, $L_{emb-align}$ not used in CMR.** As
empirically proven in Tab. 1, those losses are crucial for achieving SoTA. We disagree that they are ad-hoc: as noted
above, our representation is very different from CMR's meshes, motivating the different losses: The min-k loss densifies
the supervisory signal in landmark-less areas, while $L_{emb-align}$ fixes the coordinate distribution on the sphere.

**Reviewer 3.**   **Novelty: building on CMR.**   We solve a similar problem, but everything else is rather different from
CMR, including representation and loss functions. **Why is the model non-rigid [but] . . . rigid objects?** See lines
22–23: Since we model a class of objects, even if its instances are rigid, we still need to account for the *deformations*
*between instances* (e.g. birds deform to starling or seagull). Prior work [2,29,43,12,34,41,62] also tests the algorithms
by modelling deformations between different instances. **Combination of too many losses. Not a major concern if**
**authors apply to non-rigid objects.** CMR uses 8 loss terms in total, more than C3DM. We outperform CMR on all
datasets they use, and additionally on Freiburg Cars and FlorenceFace (all of them have non-rigid deformations). We
also demonstrate that all loss terms are crucial for good reconstruction in Tab. 1. **Handling view-dependent effects**
**with perceptual loss is unsatisfying. Use viewpoints explicitly?** In fact, C3DM *explicitly models* view-dependent
effects in the top-k loss by comparing the reference image with a *warp* of the $K$ target images produced with predicted
viewpoints. The top-k selection is instead needed to mitigate effects of *self-occlusion* (l. 182). **Limitation: relies on**
**successful NRSfM initialization.** NRSfM is used as initialization in most related methods [12,29,62]; CMR [12],
in particular, uses old rigid SfM. We don't see it as a limitation given that NRSfM from keypoints is a much easier
problem: when it fails, dense reconstruction is probably impossible. Furthermore, NRSfM supervision is injected in a
*soft* manner in (1), so can be corrected. **Why is depth prediction of a CNN considered non-parametric?** We define
non-parametric depth estimation in l. 36 and on. This is in contrast to CMR and others predicting the whole shape.

[Meta-Review · NeurIPS 2020]

The initial scores for this paper were diverging: 6: Marginally above the acceptance threshold. 5: Marginally below the acceptance threshold. 7: A good submission; accept. 4: An okay submission, but not good enough; a reject. The positive points praised by the reviewers were: - The work addresses a hard problem with an interesting and timely approach putting together existing components and including technical details to improve results. - Compelling quantitative and qualitative improvement over existing work - Reasonable ablations. The negative points: - The contributions of the work are not convincingly evaluated. - Other missing experiments testing important scenarios. - Rather ad-hoc losses and unclear motivation for parts of the model. - Some missing citations. - Missing discussion of limitations. - Several unclear points in writing. The authors provide a rebuttal, which addresses some of the weak points. This paper had an extensive and healthy post-rebuttal discussion among the positive and negative reviewers. The key points of that discussion are summarized at the end of this meta-review. As an outcome of that discussion R2 raises their score from 4 to 6. The rest of the reviewers keep their original ratings. The final scores are: 6, 5, 7, 6. However, all reviewers agree that the paper should address the following three points (listed below) before publication. This was discussed with the AC. AC has also discussed the issue with SAC. As there is no “conditional accept” decision at NeurIPS, the AC suggests (after reading the paper, consulting with the reviewers and SAC) to Accept the paper, trusting the authors that they will address these three points (listed as A.-C. below) in the final version: Summary of requested experiments for final version: A. Only replace the representation (mesh vs implicit) while keeping everything else fixed. B. Do the experiment where only 1 MLP network is used initialized with rigid-SFM output. C. Add qualitative results on PASCAL 3D+ in the paper. ###################################################### Below is the (anonymized) summary of the key exchanges among the reviewers in the post-rebuttal discussion. 1. Concerns over the lack of contribution: Similar to CMR/CSM plus AtlastNet like implicit representation?: The authors respond that being able to learn 3D shape through the texture loss is a big part of the difference why C3DM is more than CMR/CSM, as CMR/CSM did not do this. This is true -- CMR did not backprop on texture loss. However this CVPR'20 work from Henderson et al. shows that you can https://arxiv.org/abs/2004.04180. This paper may not have been known to the authors (CVPR happened around the NeurIPS deadline), so I’m fine if they correct and discuss this point in the main paper. To me, it seems that these are the main differences between CMR, CSM, and the proposed approach: (i) CMR is akin to a direct method - backpropagation through the texture results in a photometric-like loss (it’s not quite a photometric loss since a perceptual loss is used instead, but it’s close enough); (ii) CSM learns to establish correspondences from image pixels to a fixed shape template that does not adapt to the depicted shape (their articulated-CSM follow-up CVPR 2020 paper allows the template to deform, but the shape deforms based on a semi-manually defined skeleton, which does not have the capacity to capture surface details); (iii) the proposed approach learns to establish correspondences from image pixels to the parameterized surface of a (C3DPO) shape basis that then deforms to the depicted shape. In the classical debate of direct versus correspondence methods, I view the proposed method as belonging to the latter camp. My hypothesis is, similar to how correspondence methods played out in the late 90s and 2000s, the proposed approach may be less susceptible to local minima than direct methods during shape-fitting optimization. But I think there’s room to investigate this issue more fully, which may be outside the scope of this paper. Although I think (iii) is still a hybrid of CMR and CSM (but still with known keypoints). With that said, I'm changing my mind on this, I find this combination a reasonable idea. 2. Rebuttal. I'm a bit confused about the last two sentence in the first paragraph in the rebuttal. The first sentence indicates that one can learn the shape (due to reprojection) regardless of the texture quality, contradicting the first point that C3DM can learn shape through texture more. Also confused about "note that without appearance cues CMR fails to reconstruct faces". CMR on faces doesn't look good in the paper already, it's worse without the texture loss? but it contradicts their earlier point that in CMR texture is not affecting shape/pose? I also found the last two sentences in the first paragraph of the rebuttal to be unclear. We could ask that the paper clarifies the point they were trying to make in the final version. 3. Mesh vs Implicit: There is a new experience in the author response, but it seems like for training with mesh, they removed min-k percep and repro with CSM's cycle consistency and that gets worse results. Why remove min-k precep also? The right thing to do is to keep all losses equal and just use the mesh representation during training (please correct me if this is not possible and I'm not seeing it). It may point out that min-k percep is the most important loss? This is related to one of my original concern, as pointed out by other reviewers this paper looks proposes new losses and the experimental protocol does not really identify what makes this get good result (is it the representation? the new losses?). The provided ablation study does not answer this as the representation is fixed. The new results in the response is also unfortunately eludes this question. I agree with your point that they should have also tried the explicit representation with the min-k loss (I also don’t see why they couldn’t use it). This apparently important min-k loss is not much of a focal point in the paper. From the ablation study in Table 1 it does not seem to be quite important. More critically, that ablation does not offer the experiment of straight forward CSM+CMR vs their model. The number reported on the rebuttal touches on this, but it still contains the basis also -- so maybe that's the most important part? (Tab 1 in paper seems to indicate that this is so). I feel like this is quite a key information to add in order for others to build on to learn what it was that made this work well. 4. - The explicit basis is not clearly motivated. My question"However in principal can't the non-linear MLP basis learn this all" is not addressed. Atlasnet sphere should be able to also capture multiple deformations with an implicit non-linear basis function. It's more that I wanted an ablation where only one atlasnet-sphere is used instead of multiple basis. Getting AtlasNet sphere to work out of the box may be tricky due to surface self-intersection, which causes discontinuities during optimization. why do you need multiple of these implicit atlasnet-sphere like representation? Just use one implicit atlasnet-sphere MLP, instead of using a set of MLP? It's non-linear so it should be technically able to do it with one (deepSDF for ex. doesn't use multiple MLP for a single category). Table 1 removes the "basis" so it seems like this is the experiment. However there's no real discussion on that ablation study other than the caption making it hard to tell what exactly that means. If you look closely, from the text it says first term of eq(4) only, so this means that only the basis matching loss is not used, but the representation is still using multiple MLP in all experiments. 5. PASCAL3D+: I am quite concerned about introducing a benchmark on PASCAL3D+ categories (Chair and Bus) that the authors do not show any qualitative results on.